# Maternal-antenatal attachment in young pregnant women: Social support, mentors, and fear of childbirth

**Vishnupriya Alavala**[1]*, **Natacha M. De Genna**[2]

**1** College of Humanities and Sciences, Virginia Commonwealth University, Richmond, Virginia, United States of America, **2** Departments of Psychiatry, Epidemiology, Clinical and Translational Science, University of Pittsburgh School of Medicine, Pittsburgh, Pennsylvania, United States of America

* alavalav2@vcu.edu

## Abstract

The goal of this study was to examine correlates of maternal-fetal attachment in a diverse sample of young pregnant mothers. Although studies have looked at social determinants of health and maternal-fetal attachment, none have examined correlates that may be more prevalent in younger populations including prenatal drug use or fear of childbirth. We analyzed data from a prospective cohort of 154 pregnant adolescents and young adults (75.5% Black/African American, 15.6% white, 7.5% biracial, 1.4% other) ranging from 14–21 years old. Participants were recruited early in pregnancy and completed online surveys during each trimester of pregnancy. Maternal fetal attachment was measured with the Maternal Antenatal Attachment Scale (MAAS) between 30–36 weeks gestation. A linear regression model was constructed to determine the independent contributions of demographic variables (age, race, sexual minority status), maternal mental health, substance use, and social support to maternal-fetal attachment. Third trimester social support, presence of a natural mentor, not having a younger partner, and lower fear of childbirth were associated with higher MAAS scores, and therefore greater maternal fetal attachment, accounting for 32% of the variance in MAAS scores. Maternal age, substance use, and psychological symptoms were not significantly related to maternal fetal attachment in this sample. Social support and patient education to lessen fear of childbirth should be the target of interventions to improve maternal-fetal attachment in adolescent and young adult women. Fostering and supporting mentorship may also be a promising avenue to improve maternal-fetal bonding in younger patients. More research is needed to better understand the impact of perceived health, body dissatisfaction, physical fitness, healthcare access, pre-existing OB/GYN conditions, fear of breastfeeding, history of emotional neglect, and future child rearing support to address gaps in knowledge within this field.

**Data availability statement:** The data that support the findings of this study are openly available in the Dataverse data repository at https://dataverse.harvard.edu/dataset.xhtml?persistentId=doi:10.7910/DVN/WONMAT

**Funding:** Research reported in this publication was supported by the National Institute On Drug Abuse of the National Institutes of Health under Award Number R01DA046401 (NDG). The funders had no role in study design, data collection and analysis, decision to publish, or preparation of the manuscript.

**Competing interests:** The authors have declared that no competing interests exist.

## Introduction

Maternal-fetal attachment (MFA) refers to a pregnant mother's emotional connection to the growing fetus. A growing body of research has linked MFA to future outcomes for families and children. Low MFA can lead to worse maternal mood and subsequently affect postpartum bonding [1]. Lower MFA is correlated with lower prenatal maternal fetal representations, lower postnatal attachment up to one year postpartum and lower maternal self care, which can have negative implications for future parenting and parent-child bonding [1,2]. Low MFA has also been linked to issues in childhood, such as adverse behavioral, cognitive, and emotional development [2]. Higher MFA is linked to better physical neonatal outcomes (like healthy birth weight), either directly or via healthy behaviors such as abstaining from substance use, eating a balanced diet, sleeping adequately, and exercising appropriately, and preparing for their baby with pregnancy education [3]. Stronger MFA is also important for positive infant emotional development and regulation [4].

There have been varied results linking demographic characteristics to MFA, but some have found that maternal age was negatively correlated to MFA [1,5,6]. Younger mothers have been implicated to form weaker emotional attachments to the fetus, but this may not account for the interaction between maternal age and gestational age; MFA has been suggested to be higher for younger and older women later in pregnancy, and higher for middle-aged women earlier in pregnancy [6,7]. Some studies report that indicators of socioeconomic status are correlated to MFA while others do not report this association [5]. Women who were married, in a stable relationship, or living with a partner had stronger MFA in several studies [1,5]. Education level has had mixed results on its effect on MFA, possibly leading to stronger MFA from unfinished secondary to a university degree, but weaker MFA in mothers with more education [1,5,8]. Few studies found associations with employment, but it was found to be linked to stronger MFA in some studies [1,6]. Although some have investigated the mother's perception of partner support, few studies actually take into account partner age, stress, and behaviors [6].

Reproductive history and factors including pregnancy planning, pregnancy history, and prenatal substance use have also been implicated in MFA [9]. People pregnant for the first time, having an unplanned pregnancy, and previous abortions have sometimes reported lower MFA [5,6]. Mothers who had a Cesarean section or labor induction had lower MFA than mothers who had vaginal delivery in some studies [6]. Early bonding is correlated to bonding later in pregnancy and in the postpartum period [6,10]. Using substances like illicit drugs during pregnancy has been associated with low MFA and negative birth outcomes for a number of organ systems [11]. Substance abuse may contribute to impaired maternal development and increase risky behaviors, but some have found no association between drug usage (cocaine, heroin, tobacco, alcohol, etc.) and MFA [10, 11, 12].

Stress and mental health have mixed results on MFA. Depression has been associated with lower MFA, but has also been shown to be a poor predictor or unrelated to MFA [1,6]. Interestingly, MFA may predict depression later but not earlier in pregnancy. [1,6,10]. Studies have also reported mixed results for anxiety and MFA, with

higher anxiety linked to variable effects on quality and intensity of attachment [1,6,9]. Understanding and screening for the baseline state of mothers can inform professionals as to watch out for exacerbations of conditions like depression during pregnancy [6]. Higher Fear of Childbirth (FoC) has been linked to lower MFA as measured by the Maternal–Fetal Cranial Attachment Scale and Prenatal Attachment Inventory, following trends identified for other psychological disorders like anxiety [13,14]. FoC ranges from anxiety to tokophobia, or a strong sense of dread that drives women to opt for caesarean sections in some nations, potentially due to factors related to the healthcare system and culture-specific traditions [15]. More research is needed to confirm the connection between FoC and MFA using more common measures of MFA like the Maternal Antenatal Attachment Scale (MAAS) [1]. Pregnancy-related stress has been positively correlated with MFA whereas life stress has been negatively correlated with MFA, suggesting that the mental resources available for bonding with the fetus can be increased with pregnancy-related stress but taken away from external stressors [1].

The prior and current maternal environment can also play a role in MFA. Stronger social is linked to stronger MFA, although its effects on intensity and quality of MFA are still mixed in the literature [1]. Partner and interpersonal support empowers mothers and can promote better pregnancy experiences, vigilance, and resilience for mothers as they deal with current stressors and future plans. Maternal exposure to trauma and violence, whether that be from childhood or adult circumstances involving domestic violence, interpersonal violence, and sexual abuse has shown to decrease MFA [16]. Social support was found to mediate the experience of war trauma to low MFA [17]. Higher maternal wellbeing was linked to stronger MFA, although results may vary by trimester [1].

Variability in trends of MFA can be attributed to the different societal and cultural standards in place among different samples, which may affect how a pregnant woman views caring for the fetus and allocating mental resources to the pregnancy [3]. Pregnant minoritized women, those who do not have a partner, have lower socioeconomic status/income, are unemployed, or are less educated tend to receive less social support and encounter disadvantages and challenges that could lead to lower MFA as a result of societal discrimination or depletion of resources that could go towards forming a strong bond with their baby [2]. There are a dearth of studies focusing on the link between demographic factors and MFA, especially for vulnerable populations like Black/African American women, war survivors, and younger mothers [1,18]. There is an urgent need to learn more and help empower more vulnerable mothers with more targeted interventions in community settings and hospitals to improve MFA.

This study examines associations between MFA and relevant demographics and psychological factors in a diverse sample of younger people from a prospective cohort study. Although other studies have looked at the relationship between social determinants of health and pregnancy, none have surveyed a diverse cohort of young mothers on factors such as FoC, drug exposures, sleep, and partner age to discover effects on MFA. Based on prior research, we hypothesized that MFA would be lower in people with unplanned pregnancy/undesirable pregnancy timing, substance use before and/or during pregnancy, lower social support, poor mental health, poor sleep, higher FoC, and exposure to violence.

## Methods

The YoungMoms study is a prospective cohort study of pregnant women under the age of 22 enrolled from October 2019 to June 2024 from prenatal clinics in Western Pennsylvania to assess prenatal cannabis use and co-use with tobacco (R01DA046401). Recruitment started October 9, 2019. Recruitment ended June 17, 2024. Eligible patients were asked to complete an online survey and provide access to a clinical urine sample and their medical records for the pregnancy.

### Design

Obstetrics patients were recruited from prenatal clinics in Pittsburgh, PA. Written or verbal informed consent was obtained from all participants prior to data collection. After the pandemic began, consent was primarily verbal. All participants were sent a consent document to review with a research assistant. Participants were asked questions to ensure that they understood. We obtained a waiver of the requirement to obtain parental permission to participate in the study from the

 

participants who were minors (less than 18 years of age) based on the criteria in the US federal regulations at 45 CFR 46 116.(d): the research involves no more than minimal risk; the waiver will not adversely affect the rights and welfare of the subjects; the research could not practically be carried out without the waiver or alteration and whenever appropriate, the subjects will be provided with additional pertinent information. Participants who completed the baseline survey during the first trimester were recruited into the longitudinal study and assessed again via online survey, telephone interviews, and clinical urine samples during their second (20–26 weeks) and third (30+weeks gestation) trimesters. This study was approved by the University of Pittsburgh IRB STUDY18100150.

## Participants

At target prenatal clinics, English-speaking pregnant patients younger than 22 years old were eligible for recruitment. Study materials were in English, so non-English speaking patients were excluded from the study. Patients in treatment for opioid use disorder were excluded. Patients who gave birth prior to completing the baseline survey were ineligible for the study. If a patient gave birth prior to completing the third trimester survey, they were not withdrawn from the study, but had missing data at that time point.

Recruitment for the study was paused between March 2020 and June 2020 due to the COVID-19 pandemic. The longitudinal study recruitment involved participants who completed the first trimester baseline survey, were at < 14 weeks gestation, and had viable pregnancies. The target population of obstetrics patients of specific clinics were successfully recruited via phone, text, and/or email. Participants who completed the MAAS measure did not differ from the baseline sample in age, race, childhood SES, school enrollment, employment status, sexual minority status, or presence of a natural mentor.

## Measures

**Maternal antenatal attachment scale.** MFA was measured with a commonly used instrument, the Maternal Antenatal Attachment Scale (MAAS), relevant to delineating the mother's behaviours, feelings, and attitudes to the fetus [6,19]. The MAAS includes 19 items assessing emotional attachment such as being aware of the baby, wanting to keep the baby safe, amount of time spent thinking about the baby, and looking forward to meeting the baby at delivery [20,21]. Different items have been removed for prior studies, and in this study, this item was removed to avoid traumatizing participants, given their young age: "If the pregnancy was lost at this time (due to miscarriage or other accidental event) without any pain or injury to myself, I expect I would feel:" was omitted after consideration. Exact questions and available responses can be found in S1 Table. Cronbach's alpha value for the 18-item version of the MAAS used in the YoungMoms study was 0.767, indicating acceptable internal consistency, as seen in the original instrument [20,21].

The total MAAS score was used over subscale scores that have had inconsistent reliability estimates [22]. Higher MAAS scores indicate stronger MFA [23].

**Fear of Childbirth (FoC).** The Revised Wijma Delivery Expectancy/Experience Scale (WDEQ-R) was used to measure FoC [24]. Although the WDEQ-R had 11 items, two items about maternal and child death were removed for this study. The adapted measure with 9 items was used in the third trimester survey (Cronbach's α=0.9). This measure is scored such that mothers with less FoC have higher scores.

**Demographic information.** Age was abstracted from the medical chart, and other demographic data were reported by mothers in the baseline survey. including race, ethnicity, gender, educational status, employment, and relationship status. To assess ethnicity, mothers were asked "Are you Hispanic/Latina?" with a yes/no forced choice response. Race was measured by asking mothers to identify according to the U.S. Office of Management and Budget (OMB) categories: "How would you describe your race (select one or more responses)?" with the following response options: Black/African American, White, Biracial, Asian, Other (specify). Sexual orientation was assessed using the question from the 2018

Pittsburgh Public Schools Youth Risk Behavior Survey, which was adapted from the Centers for Disease Control and Prevention's Youth Risk Behavior Surveillance System (YRBSS). The measure was adapted for the survey so that participants could indicate more than one choice and new responses options were added (e.g., mostly heterosexual/mostly straight, queer, asexual). In the third trimester survey, Participants were again asked to indicate their relationship status. A dichotomous variable was created for partner age (0 = same age or older partner, 1 = younger partner) for participants reporting being in a relationship. Before conducting analyses on the analytic sample, missing responses were coded as not having a younger partner, increasing the sample size and power of this variable.

**Natural mentors.**  Availability of a natural mentor was assessed in the baseline survey with the following question, "Is there an adult 25 years or older who is not related to you who made an important, positive difference in your life?" The following responses were counted as natural mentors: someone from my church, Big Sister program, someone from another program, teacher, coach, someone from my neighborhood, and someone else (specify). Romantic partners were not coded natural mentors.

**Pregnancy intendedness, desire, and timing.**  The 6-item London Measure of Unplanned Pregnancy (LMUP) measured these constructs in the baseline survey. Language was adapted for a U.S. population (e.g., please "choose" the statement rather than please "tick" the statement) [25].

**Stress.**  Stress experienced in the past month was measured using the short form of the Perceived Stress Scale (PSS-4) [26]. This 4-item scale has demonstrated reliability and validity in pregnant populations [27]. Responses ranged from 0 (never) to 5 (very often) for each of the 4 items.

**Social support.**  The following three items from the 2018 Pittsburgh Public Schools YRBSS were used to assess social support available to the mother in each trimester on a 5-point Likert scale: "Is there someone you really count on to be dependable when you need help?"; "Is there someone you really count on to care about you, regardless of what is happening to you?"; and "Is there someone you really count on to help you feel better when you are feeling generally down-in-the-dumps?" [28]

**Depressive and anxiety symptoms.**  Recent depressive symptoms were measured in each trimester with the 8-item Patient-Reported Outcomes Measurement Information System (PROMIS) emotional distress scale [29]. Recent symptoms of anxiety were assessed using the 4-item PROMIS anxiety scale later in pregnancy.

**Sleep.**  The Pittsburgh Sleep Quality Index (PSQI) was used to measure sleep duration and quality in the second and third trimester [30]. The total sleep score was used in these analyses. Higher values correspond to worse sleep quality [31].

**Exposure to violence.**  A short form of the Child Trauma Questionnaire was used to measure exposure to child abuse and neglect (CTQ) [32]. Bullying was assessed on a 5-point Likert scale for frequency of being bullied ranged from never to always with the following question from the YRBSS: "How often are/were you bullied at school?" Intimate partner violence (IPV) was assessed with 3 items modified from the Conflict Tactics Scale-2 (CTS-2) and the Sexual Experiences Survey to measure lifetime and recent (past 3-month) experiences with physical and sexual abuse by a partner [33,34]. A dichotomous variable for lifetime IPV (0 = no IPV, 1 = IPV) was used in the analysis.

**Experiences of discrimination.**  Nine questions in the survey assessed common daily experiences of discrimination [35,36]. Responses were summed for this analysis, with greater scores indicating more experiences of discrimination.

**Substance use.**  Mothers were asked about the frequency of substance use before and during pregnancy in the baseline survey. During the second and third trimesters, participants were asked if they had used tobacco products in the past 2 weeks, any e-cigarettes since the previous trimester, any cannabis products in the past month, and any prescription drugs for non-medical purposes or illicit drugs since they became pregnant.

**Time period.**  As the effects of the COVID pandemic may have impacted MFA, we included a variable for the time period when mother joined the study. Data collection time was represented using the following dummy codes: 0 = October 2019-March 2020, 1 = June 2020-May 2021, 2 = June 2021-May 2022, 3 = June 2022-May 2023, 4 = June 2023-June 2024.

## Analysis

Quantitative variables were handled in the analyses. Most variables were treated as nominal or continuous variables, and there was no manipulation of binning variables for analysis. Age was treated as a continuous variable instead of a discrete variable. Different sets of variables were tested using bivariate analysis to determine significant associations (threshold for significance at 0.05) with MAAS and if they followed trends established in the literature. Linear regression was used to determine independent contributions of demographic variables (age, race, sexual minority status), maternal mental health, substance use, and social support to MAAS scores. Homoscedasticity was present when reviewing a standardized residual regression plot. There was no evidence of severe multicollinearity between variables in the hierarchical regression model. All analyses were conducted using IBM SPSS Statistics software (Version 29.0.2.0 [37].

Participants who had missing MAAS scores from their third trimester survey were not included in these analyses. For participants that completed at least 4 of the items in the PROMIS depression screener, mean replacement was used for any missing data. Missing values for all other variables were treated as missing. All non-missing data was used, with valid percentages reported after accounting for missing values. In bivariate correlation analysis using the Pearson correlation coefficient, cases were excluded pairwise. In establishing significant correlations between MAAS and factors of interest, independent sample 2-sided t-tests (between-subjects) were conducted following Levene's Test for equal variances or One-Way ANOVAs followed by Tukey post hoc testing for categorical variables. Cases of missing values were excluded analysis by analysis. In linear regression analysis, missing cases were excluded listwise in the model (n = 142).

## Results

Fig 1 describes baseline survey and third trimester survey recruitment. Of the 975 eligible patients, 219 declined interest in the study before speaking with a research assistant, 210 declined after speaking with a research assistant, eight provided consent but left their surveys incomplete, and one consented but completed the survey after pregnancy termination. Thirty-two individuals became ineligible for the study before survey completion because of termination, pregnancy loss, or early delivery. One patient's survey responses were unrecoverable because of technical issues with the online survey. 504 patients completed the first trimester baseline survey. These patients were contacted to participate in the longitudinal study. Longitudinal survey outreach was conducted virtually during the COVID-19 pandemic. 188 obstetrics patients of the target clinics were successfully recruited, 50 did not respond, and 23 declined to participate in the longitudinal study. In the longitudinal study, 153 completed the 2nd trimester survey and 154 completed the third trimester survey. 142 patients with non-missing MAAS scores during follow-up surveys were analyzed.

Table 1 shows demographic characteristics of the larger sample of 154 individuals who completed the third semester survey and the analytic sample who had nonmissing MAAS values. Missing responses for the Younger Partner (n = 109) variable were recoded to represent not having a younger partner, reaching n = 142 in the analytic sample. Most participants (61.9%) report having a high school diploma as their highest level of education, with only one participant having completed four years of college.

Third trimester survey variables had n = 147 except the following: Employed (n = 146), Sexual minority (n = 146), and Younger partner (n = 109). The analytic sample had n = 142 except "Employed" which had n = 141. SD = standard deviation

Descriptive statistics for continuous variables are shown in Table 2.

We examined drug use as a function of MFA: rates of drug use before and during pregnancy are presented in Table 3.

There were no missing scores of 142 respondents with nonmissing MAAS scores for the substance use outcomes in Table 3.

Other pregnancy characteristics like pregnancy desirability, intendedness, and timing in addition to experiences of violence are shown in Table 4.

Bivariate correlations were calculated between MAAS and continuous variables of interest using Pearson correlation. As FoC scores increased (indicating less fear), MAAS scores increased, as shown in Table 5. Similarly, social support

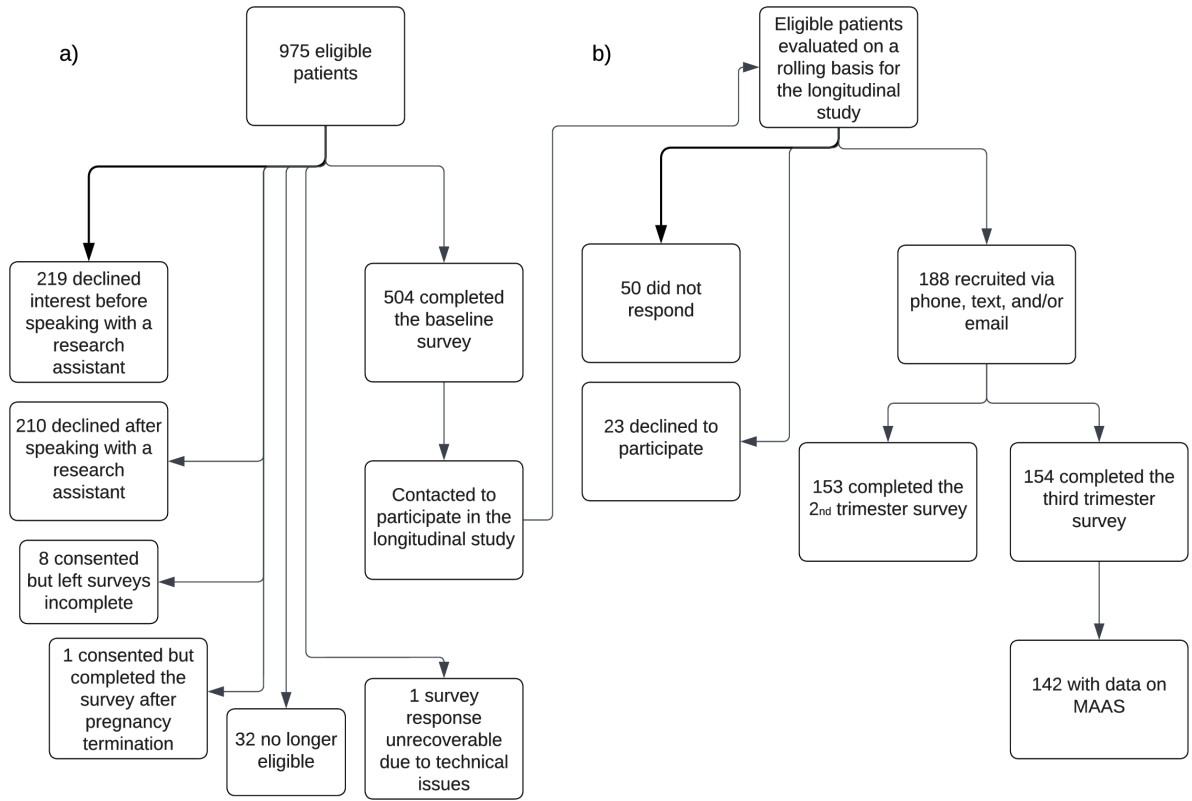

**Fig 1. Numbers of individuals at each stage of study.** a) Baseline survey recruitment. 975 potentially eligible patients from target prenatal clinics were all English-speaking pregnant patients younger than 22 years old, not in treatment for opioid use disorder, and had a viable pregnancy. Patients were examined for eligibility, confirmed eligible, and included in the study, with reasons for non-participation at each stage. b) Longitudinal survey recruitment. Outreach was conducted virtually during the COVID-19 pandemic. 142 patients with non-missing MAAS scores during follow-up surveys were analyzed.

and MAAS scores were also positively correlated. As third trimester depressive symptoms and stress decreased, MAAS scores increased. As third trimester PSQI sleep scores (and subscores of sleep duration and sleep medications) decreased, signifying higher sleep quality, MAAS scores increased.

The only significant result from the t-tests on substance use and MFA was for co-use of nicotine and cannabis during the third trimester, as seen in Table 6.

Participants reporting "right time" vs "wrong time" for pregnancy intendedness had significantly different MAAS scores at 77.78 and 71.20, respectively as shown in Table 7 (p = .025). Those reporting five or more experiences of discrimination had lower MAAS scores than those who did not report many experiences of discrimination at 71.56 and 76.45, respectively (p = .027). Those with a younger partner reported lower MAAS scores than those who did not at 70.46 and 75.84, respectively (p = .041). Those with vs without a natural mentor reported more favorable MAAS scores at 77.58 and 74.57, respectively (p = .041).

A linear regression was constructed to establish correlates of MAAS using three different blocks with the statistically significant variables from the bivariate analyses (Tables 5 and 7). FoC, third trimester social support, natural mentor, and younger partner were the only significant correlates of MAAS scores in the final model. As seen in Table 8, more third trimester social support (Beta = .343, t = 4.224, p < .001), having a natural mentor (Beta = .191, t = 2.674, p = .008), not having

**Table 1. Characteristics of study participants and analytic sample characteristics.**

| Sample Characteristics | Third trimester participants (n = 147) | Analytic sample (n = 142) |
|---|---|---|
| Age | Mean = 19.71 (SD = 1.61) | Mean = 19.70 (SD = 1.60) |
| Race | | |
| Black/African-American | 75.5% | 74.6% |
| White | 15.6% | 16.2% |
| Biracial | 7.5% | 7.7% |
| Asian American | 1.4% | 1.4% |
| Latine ethnicity | 2.7% | 2.8% |
| Gender | | |
| Cisgender | 98.6% | 98.6% |
| Non-binary | 1.4% | 1.4% |
| Bisexual | 20.4% | 21.1% |
| Plurisexual | 30.6% | 31.7% |
| Employed | 43.8% | 43.3% |
| Full-time | 22.6% | 22% |
| Part-time | 21.2% | 21.3% |
| Enrolled in school | 29.9% | 29.6% |
| Marital status | | |
| Single, not dating | 24.5% | 25.4% |
| Single, dating someone | 39.5% | 38.7% |
| Living with a partner | 29.9% | 29.6% |
| Married | 4.1% | 4.2% |
| Separated, dating | 0.7% | .7% |
| Widowed, not dating | 0.7% | .7% |
| Missing/did not answer | 0.7% | .7% |
| Sexual minority | 39% | 38.7% |
| Younger partner | 11.9% | 9.2% |
| Experienced bullying | 50.3% | 49.3% |

**Table 2. Continuous variables in analytic sample.**

| Variable of Interest | Sample size | Mean | Standard Deviation | Cronbach's Alpha | Range | |
|---|---|---|---|---|---|---|
| | | | | | Potential | Calculated |
| MAAS (18-item) | 142 | 75.3451 | 8.35816 | 0.8 | 19–90 | 49-90 |
| Fear of childbirth | 137 | 72.33 | 10.733 | 0.9 | 16-96 | 35-96 |
| First trimester social support | 139 | 10.47 | 3.437 | 0.9 | 3-12 | 3-16 |
| First trimester perceived stress | 138 | 9.49 | 3.137 | 0.2 | 4-20 | 4-17 |
| First trimester depressive symptoms | 141 | 16.06 | 7.574 | 0.9 | 8-40 | 8-40 |
| Third trimester Social Support | 142 | 9.77 | 2.638 | 0.9 | 3-12 | 3-12 |
| Third trimester perceived stress | 141 | 9.99 | 3.172 | 0.4 | 4-20 | 4-18 |
| Third trimester depressive symptoms | 142 | 15.08 | 7.469 | 0.9 | 8-40 | 8-40 |
| Third trimester anxiety | 142 | 7.51 | 3.522 | 0.8 | 4-20 | 4-20 |
| Third trimester PSQI Total Sleep Score | 136 | 8.46 | 3.388 | 0.6 | 0-19 | 1-19 |

**Table 3. Substance use in analytic sample.**

| Variable of Interest | Percent |
| --- | --- |
| Lifetime illicit drug use | 12.7% |
| Illicit drug use during first trimester of pregnancy | 0% |
| Tobacco use before pregnancy (self-report) | 25.4% |
| Nicotine use before pregnancy (self-report) | 49.3% |
| Marijuana use before pregnancy (self-report and urine) | 43.7% |
| Tobacco use in first trimester (self-report) | 12.7% |
| Nicotine use in first trimester (self-report) | 14.8% |
| Marijuana use in last month from third trimester survey (self-report) | 19.0% |
| Tobacco use in past two weeks from third trimester survey (self-report) | 4.9% |
| Vaping during the third trimester | 4.2% |
| Significant other vaping | 12.7% |
| Significant other smokes cigarettes | 20.4% |
| Significant other smokes marijuana | 35.9% |
| Significant other vapes marijuana | 7.7% |
| Significant other binge drinks | 1.4% |
| Marijuana use during third trimester (self-report and urine) | 21.1% |
| Nicotine use during third trimester (self-report and urine) | 23.9% |
| Co-use of nicotine and cannabis | 12.7% |

**Table 4. Other categorical variables in analytic sample.**

| Variable of Interest | Percent | Number of respondents |
| --- | --- | --- |
| Pregnancy desired in first trimester | | 140 |
| Wanted to have a baby | 33.6% | |
| Mixed feelings about having a baby | 39.3% | |
| Did not want to have a baby | 27.1% | |
| Pregnancy intendedness in first trimester | | 141 |
| Wanted to get pregnant | 24.8% | |
| Mixed feelings | 25.5% | |
| Did not mean to get pregnant | 49.6% | |
| Pregnancy timing in first trimester | | 141 |
| Right time | 29.1% | |
| Ok, but not best time | 60.3% | |
| Wrong time | 10.6% | |
| Partner pregnancy intentions | | 139 |
| Agreed on being pregnant | 26.6% | |
| Talked about it but hadn't agreed on pregnancy | 55.4% | |
| Never talked about having a kid together | 18.0% | |
| Lifetime IPV measured in first trimester | 19.1% | 141 |
| Recent IPV in first trimester | 5.6% | 142 |
| Many experiences of discrimination (>=5) | 22.5% | 142 |
| Natural mentor | 35.3% | 136 |

**Table 5. Bivariate correlations with maternal-antenatal attachment scale scores in analytic sample.**

| Variable of Interest | Sample size | Correlation Coefficient | Significance (p-value) |
|---|---|---|---|
| Fear of childbirth | 137 | .396** | <.001 |
| First trimester Social Support | 139 | .230* | .006 |
| First trimester perceived stress | 138 | −.165 | .053 |
| First trimester depressive symptoms | 141 | −.156 | .065 |
| Third trimester Social Support | 142 | .486** | <.001 |
| Third trimester perceived stress | 141 | −.234* | .005 |
| Third trimester depressive symptoms | 142 | −.199* | .018 |
| Third trimester anxiety | 142 | −.104 | .219 |
| Third trimester PSQI Sleep Duration | 142 | −.349** | <.001 |
| Third trimester PSQI Sleep Meds | 141 | −.221* | .009 |
| Third trimester PSQI Total Sleep Score | 136 | −.293** | <.001 |

*p < .05,

**p < .001; cases of missing values were excluded pairwise. Significance values were 2-tailed.

**Table 6. Bivariate Associations Between Maternal-Antenatal Attachment Scale Scores and Drug Use in Analytic Sample.**

| Variable of Interest | Student's t-test, t (df) | Significance (p-value) |
|---|---|---|
| Lifetime Illicit Drug Use | .036 (140) | .971 |
| Tobacco use before pregnancy (self-report) | −1.076 (140) | .284 |
| Nicotine use before pregnancy (self-report) | .404 (140) | .687 |
| Marijuana use before pregnancy (self-report and urine) | .654 (140) | .514 |
| First trimester tobacco use (self-report) | −.989 (140) | .324 |
| First trimester nicotine use (self-report) | −.134 (140) | .894 |
| Third trimester marijuana use (self-report) | .929 (140) | .355 |
| Third trimester cigarette use (self-report) | 1.180 (140) | .240 |
| Third trimester vaping | −.046 (140) | .963 |
| Third trimester significant other vaping | .972 (140) | .333 |
| Third trimester significant other smokes cigarettes | −.099 (140) | .921 |
| *Third trimester significant other smokes marijuana* | −.399 (132.572) | .690 |
| *Third trimester significant other vapes marijuana* | −.808 (20.413) | .428 |
| *Third trimester significant other binge-drinking* | .317 (1.004) | .805 |
| Third trimester marijuana use (self-report and urine) | .795 (140) | .428 |
| Third trimester nicotine use (self-report and urine) | −.711 (140) | .478 |
| *Third trimester co-use of nicotine and cannabis* | −2.683 (32.926) | .011* |

*p < .05; number of respondents in all cases was 142 for all variables except co-use of nicotine and cannabis where n = 112. Independent sample 2-sided t-tests (between-subjects) were conducted. Italicized variables indicate situations where the equal variances could not be assumed as established by Levene's Test (p < 0.05), so the adjusted p-value is reported. Cases of missing values were excluded analysis by analysis.

a younger partner (Beta = −.209, t = −2.944, p = .004), and lower FoC (Beta = .192, t = 2.183, p = 0.031) were related to higher MFA in this sample, accounting for 32% of variance in MAAS scores.

## Discussion

This study investigated maternal-fetal attachment among younger pregnant people. Third trimester social support, presence of a natural mentor, not having a younger partner, and less fear of childbirth were related to MFA in this population.

**Table 7.  Bivariate Associations Between Maternal-Antenatal Attachment Scale Scores and Other Covariates in Analytic Sample.**

| Variable of Interest | Significance Test | Significance (p-value) | Number of respondents |
|---|---|---|---|
| Pregnancy desire | F (total df) =.844 (139) | .432 | 140 |
| Pregnancy intendedness | F (total df) =.343 (140) | .710 | 141 |
| Pregnancy timing | F (total df) = 3.803 (140) | .025* | 141 |
| Partner pregnancy intentions | F (total df) = 5.230 (138) | .006* | 139 |
| *Lifetime IPV measured in first trimester* | t (df) =.178 (33.166) | .860 | 141 |
| *Recent IPV in first trimester* | t (df) = 1.232 (7.324) | .256 | 142 |
| *Many experiences of discrimination (>=5)* | t (df) = 2.305 (37.927) | .027* | 142 |
| Younger partner | t (df) = 2.242 (140) | .027* | 142 |
| Natural mentor | t (df) = −2.068 (134) | .041* | 136 |

*p < .05; One-Way ANOVAs for pregnancy desiredness, intendedness, and timing in the first trimester were performed followed by Tukey post hoc testing. Independent samples 2-sided t-tests (between-subjects) were performed for the remaining binary variables. Italicized variables indicate situations where the equal variances could not be assumed as established by Levene's Test (p < 0.05), so the adjusted p-value is reported. Cases of missing values were excluded analysis by analysis.

**Table 8.  Regression Models on Abbreviated Maternal-Antenatal Attachment Scale Scores in Analytic Sample.**

| n = 142 | Standardized Coefficients Beta | t-value | Significance (p-value) | Variance inflation factor (VIF) |
|---|---|---|---|---|
| Third trimester social support | 0.446 | 5.725** | <.001 | 1.124 |
| Third trimester PSQI total sleep | −0.120 | −1.542 | .125 | 1.124 |
| Third trimester social support | 0.444 | 5.881** | <.001 | 1.125 |
| Third trimester PSQI total sleep | −0.103 | −1.356 | .177 | 1.131 |
| Natural mentor | 0.156 | 2.171* | .032 | 1.016 |
| Younger partner | −0.198 | −2.763* | .007 | 1.010 |
| Third trimester social support | 0.343 | 4.224** | <.001 | 1.358 |
| Third trimester PSQI total sleep | −0.042 | −0.524 | .601 | 1.328 |
| Natural mentor | 0.191 | 2.674* | .008 | 1.049 |
| Younger partner | −0.209 | −2.944* | .004 | 1.041 |
| Fear of childbirth | 0.192 | 2.183* | .031 | 1.590 |
| First trimester pregnancy timing | −0.114 | −1.516 | .132 | 1.171 |
| Third trimester dual use of nicotine and cannabis | −0.004 | −0.063 | .950 | 1.018 |

*p < .05,

**p < .001; cases of missing values were replaced with the mean.

The findings on social support and fear of childbirth were consistent with prior studies using older populations finding that MFA was greater among pregnant women reporting more social support and less fear of childbirth [1,13,14]. No prior research has examined the role of natural mentors or partner age in MFA. However, having a natural mentor was related to better mental health in one study of Black adolescent mothers [38]. Mentorship may allow for a stronger support system available for the younger pregnant woman to rely on while navigating dynamic changes to her lifestyle and body. Similarly, this is the first study looking at the correlation between parnter age and young mother's bond with the baby. Very young mothers with younger partners may not be as equipped to handle stresses of pregnancy, leading to decreased MFA and adverse downstream effects, so these couples may need greater support and education. With more studies revealing the effects of prenatal bonding on postnatal bonding, socioemotional, and behavioral outcomes, this research helps fill the gap in research on younger pregnant people in the US [39].

Initially presented by Condon (1993), the MAAS looked to distinguish the mother's emotional connection to the fetus from the mother's feelings to pregnancy [19]. Condon (1993) hypothesized that mothers with low MAAS scores were more likely to use drugs and have future insecure attachment to their child [19]. Substance use was not associated with MAAS scores in this study. This inconsistency may be attributed to sample differences, as this study includes very young mothers with much higher rates of substance use compared to the general public [40–42]. In a clinical context, the MAAS has been used as a reliable tool to understand the mother's psychological health, and serve as a signal for interventions like therapy to promote stronger bonding. Lower scores are concerning for struggling mothers who may need support to overcome emotional detachment, depression, and/or anxiety during pregnancy. Other researchers have used MAAS to understand outcomes in pregnancy such as infant parent-child interactions, child development, and breastfeeding [4,43,44].

MFA was shown to be positively associated with greater third trimester social support, corroborating ideas that a strong social support system for pregnant women is conducive to strengthening the maternal-fetal bond. Peer support throughout pregnancy may specifically help women from marginalized communities [45]. Social support in the first trimester helps quell initial feelings of distress for pregnant women, but as it is early in the pregnancy, it may not be as pivotal to MFA as support later on as the woman is preparing for pregnancy and accommodating her changing body. Social support in the third trimester may help mothers feel reduced anxiety during the final stages of pregnancy, and therefore be empowered to continue caring for themselves [46]. Researchers have found that mothers with support, and thereby relationship security and satisfaction, from their significant other mid-pregnancy improves expectant mother's psychological state and infant distress (Stapleton et al., 2014). Partner support has been shown to lower risk of negative mental health and prenatal drug use [47] and has also positive implications for socioemotional development of the child [44]. Greater partner support may dissuade low moods and inspire empowerment for women to continue caring for themselves during their pregnancy, especially as young pregnant women are at risk for mental health concerns [48]. Therefore, such positive influence may indirectly promote a healthier lifestyle for women and lead to better pregnancy outcomes. As social support during the end of pregnancy was most strongly associated with better MFA in this study, there may be utility in establishing peer support groups and referrals in the clinical setting to strengthen partner support and help younger mothers during this challenging period of pregnancy.

The literature on natural mentors defines them as older adults who are not parents or romantic partners who can provide additional guidance to young people, which may be especially important for pregnant Black and Hispanic adolescents [38]. Hurd and Zimmerman (2010) found that natural mentoring moderated the effects of stress on depressive or anxiety symptoms in pregnant Black adolescents. These mentors may play a role in providing intangible resources like emotional guidance, serve as role models for healthy coping behaviors, promote positive self-image for young pregnant women, and encourage an atmosphere of inclusion in a situation vulnerable to stigma among other positive effects [38]. Similarly, our study of a diverse sample of young pregnant women contributes to prior findings by supporting stronger MFA for those reporting a natural mentor. Natural mentorship may be a valuable tool to help young pregnant women stay resilient and protect against development of psychological symptoms of stress, anxiety, and depression. Offering avenues for formal mentorship or environments for mentors and mentees to bond and reflect is promising and should be normalized as helpful interventions for these populations.

This study also highlighted the importance of fear of childbirth (FoC) for MFA. Previous studies have shown that increased fear of childbirth was associated with decreased MFA in samples with an older average age [13,14]. In one US study, FoC was more common among women who were Black women, lower income, had less education, high-risk pregnancies, pre existing conditions, suffered from prenatal depression, and were surveyed later in pregnancy [49]. COVID-19 pandemic-related concerns also increased the probability of that study, showing that external stressors and potential for adverse health outcomes for their baby exacerbated their anxiety about childbirth [49]. There is likely a bidirectional relationship between MFA and FoC, as other studies have reported that as MFA increases, perceptions of traumatic birth decreased, thereby likely decreasing FoC [20]. FoC is not only associated with worse experiences with childbirth, but

also with worse rates of breastfeeding and postpartum mental health and thus may also affect postnatal maternal-infant attachment [50,51]. Interventions such as cognitive/cognitive behavioral therapy, therapeutic conversations, education on pregnancy, midwife and doula care, and mindful exercise may be worth implementing in these instances [52]. However, more research and methodology should be established for these interventions to test whether they are truly effective in reducing fear of childbirth and increasing MFA.

Several factors associated with MFA in the literature were related to MAAS score in bivariate analysis but not significant in the regression model. For example, mental health has been linked to better MFA in prior work [1,2,6,9,53]. The significant correlation with sleep quality is consistent with research regarding positive lifestyle changes on MAAS so pregnant women reporting higher quality, better sleep are expected to report higher MFA [3]. Other studies have found that pregnancy planning/intendedness is related to higher MFA but in this study of younger mothers with mostly unplanned and unintended pregnancies, perceptions of pregnancy timing seemed more important for MFA [20]. We also examined the role of partner preparedness for pregnancy and partner substance use, but these partner characteristics were not related to MFA in this study. First trimester social support was also not significant in the final regression model. MFA may change with gestational age, so studies may underestimate MFA if mothers are surveyed too early in the pregnancy [6]. Overall, the results of this study suggest that these factors may be less important for MFA in younger mothers after considering the effects of social support, natural mentors, partner age, and fear of childbirth.

Much of the variability in studies on MFA may be related to differences in measures and samples. There are over 12 different measures of MFA, including the MAAS, and some studies examine separate dimensions of MFA, such as "intensity" (strength of mother's preoccupation) and "quality" (closeness of relationship) components [1]. The total MAAS score was used in this study because it has been shown to have better reliability than the subscale scores [54]. Many studies have used an abbreviated version of the original MAAS [22]. For example, a version of the MAAS developed for the Polish population did not include items involving the mother's diet or talking to the baby when the mother was alone [22]. In the current study, a question regarding stillbirth was omitted to avoid traumatizing participants, some of whom were as young as 14 and 15 years of age. Nonetheless, the average score for the sample indicates good MFA and is similar to that reported in other studies [20]. Other differences in correlates of MFA in the literature may be related to variability in samples. Some participants are excluded from research on MFA because of language barriers, social determinants of health that prevent regular follow-ups, and loss of participants in longitudinal cohort studies. Alhusen and others have called for more research using diverse samples [11].

### Limitations

Although this study had many strengths including a prospective cohort of pregnant adolescent and young adults with measures of many factors previously associated with MFA throughout pregnancy, there are some important limitations to consider. The study design was longitudinal but MFA was only measured once and this was a cross-sectional analysis. Responses on the MAAS may change based on quickening or an individual's specific situation, such as individuals having more ideation of the baby when they are able to access ultrasounds [55,56]. The correlational design poses limitations to establishing causality, so it is impossible to determine if social support and fear of childbirth preceded or were the result of poorer MFA.

In future research, MFA should be measured in each trimester to adjust for changes during the pregnancy, to determine if these changes are related to risk and protective factors that may help identify targets for intervention during different trimesters. Many participants who were not in a relationship may have not provided a response for the question as to whether they had a younger partner, explaining the lower number of respondents in Table 1. This limitation was addressed by recoding the missing responses to not having a younger partner, increasing the power of the variable.

Choice and use of measures were another important limitation. Some of the measures in the current study, including the instruments for MFA and FoC, were adapted for the younger sample, which limits comparison to other studies. Other

constructs related to MFA in the literature were not measured, including knowledge of fetal sex, physical fitness, and body satisfaction [6,23,57]. Finally, there may have been response bias from self-reporting and social desirability bias, especially as participants were asked sensitive questions about substance use and negative feelings toward their unborn child. However, surveys were administered online after assuring participants of confidentiality.

## Conclusions

The results of this study confirm the importance of social support for maternal-fetal attachment in younger pregnant women, and highlight the unique contribution of mentorship for this population. In addition to enhancing social support and mentorship opportunities, findings suggest that interventions to reduce fear of childbirth may strengthen maternal-fetal attachment in younger mothers. Further longitudinal research should investigate the role of body image, religion, knowledge of fetal sex, physical fitness and other health behaviors to better understand the development of MFA and how to best strengthen it.

## Supporting information

**S1 Table. MAAS items with response options.**
(DOCX)

## Acknowledgments

We would like to acknowledge all the participants who offered their support with the study so that we could learn more about maternal-fetal attachment in younger mothers. Thank you to Mr. John Cyrus, a Research and Education Librarian at Virginia Commonwealth University, for helping edit the manuscript.

## Author contributions

**Conceptualization:** Natacha M. De Genna.

**Data curation:** Natacha M. De Genna.

**Formal analysis:** Vishnupriya Alavala.

**Funding acquisition:** Natacha M. De Genna.

**Investigation:** Vishnupriya Alavala, Natacha M. De Genna.

**Methodology:** Natacha M. De Genna.

**Software:** Vishnupriya Alavala.

**Visualization:** Vishnupriya Alavala.

**Writing – original draft:** Vishnupriya Alavala.

**Writing – review & editing:** Natacha M. De Genna.

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
