## [Decision Letter · Decision Letter 0]

Dear Dr. Alavala,

Thank you for submitting your manuscript to PLOS ONE. After careful consideration, we feel that it has merit but does not fully meet PLOS ONE’s publication criteria as it currently stands. Therefore, we invite you to submit a revised version of the manuscript that addresses the points raised during the review process.

**Unfortunately, I had great difficulty finding suitable reviewers for this manuscript. Ten review candidates declined and another ten did not respond to the invitation at all. In order not to prolong the review process any further, I have now decided to settle for a single review. You will see that it is very short and positive. I would therefore ask you to respond briefly to the proposal made in the report. Then I will probably accept the manuscript without sending it to external reviewers again.**

We look forward to receiving your revised manuscript.

Kind regards,

Wolfgang Blenau

Academic Editor

PLOS ONE

**Journal Requirements:**

1. When submitting your revision, we need you to address these additional requirements. Please ensure that your manuscript meets PLOS ONE's style requirements, including those for file naming. The PLOS ONE style templates can be found at https://journals.plos.org/plosone/s/file?id=wjVg/PLOSOne_formatting_sample_main_body.pdf and https://journals.plos.org/plosone/s/file?id=ba62/PLOSOne_formatting_sample_title_authors_affiliations.pdf 2. Thank you for stating in your Funding Statement: Research reported in this publication was supported by the National Institute On Drug Abuse of the National Institutes of Health under Award Number R01DA046401 (NDG). The funders had no role in study design, data collection and analysis, decision to publish, or preparation of the manuscript.  Please provide an amended statement that declares *all* the funding or sources of support (whether external or internal to your organization) received during this study, as detailed online in our guide for authors at http://journals.plos.org/plosone/s/submit-now.  Please also include the statement “There was no additional external funding received for this study.” in your updated Funding Statement. Please include your amended Funding Statement within your cover letter. We will change the online submission form on your behalf. 3. We note that this data set consists of interview transcripts. Can you please confirm that all participants gave consent for interview transcript to be published? If they DID provide consent for these transcripts to be published, please also confirm that the transcripts do not contain any potentially identifying information (or let us know if the participants consented to having their personal details published and made publicly available). We consider the following details to be identifying information:- Names, nicknames, and initials- Age more specific than round numbers- GPS coordinates, physical addresses, IP addresses, email addresses- Information in small sample sizes (e.g. 40 students from X class in X year at X university)- Specific dates (e.g. visit dates, interview dates)- ID numbers Or, if the participants DID NOT provide consent for these transcripts to be published:- Provide a de-identified version of the data or excerpts of interview responses- Provide information regarding how these transcripts can be accessed by researchers who meet the criteria for access to confidential data, including:a) the grounds for restrictionb) the name of the ethics committee, Institutional Review Board, or third-party organization that is imposing sharing restrictions on the datac) a non-author, institutional point of contact that is able to field data access queries, in the interest of maintaining long-term data accessibility.d) Any relevant data set names, URLs, DOIs, etc. that an independent researcher would need in order to request your minimal data set. For further information on sharing data that contains sensitive participant information, please see: https://journals.plos.org/plosone/s/data-availability#loc-human-research-participant-data-and-other-sensitive-data If there are ethical, legal, or third-party restrictions upon your dataset, you must provide all of the following details (https://journals.plos.org/plosone/s/data-availability#loc-acceptable-data-access-restrictions):a) A complete description of the datasetb) The nature of the restrictions upon the data (ethical, legal, or owned by a third party) and the reasoning behind themc) The full name of the body imposing the restrictions upon your dataset (ethics committee, institution, data access committee, etc)d) If the data are owned by a third party, confirmation of whether the authors received any special privileges in accessing the data that other researchers would not havee) Direct, non-author contact information (preferably email) for the body imposing the restrictions upon the data, to which data access requests can be sent 4. Please include captions for your Supporting Information files at the end of your manuscript, and update any in-text citations to match accordingly. Please see our Supporting Information guidelines for more information: http://journals.plos.org/plosone/s/supporting-information.

Reviewers' comments:

Reviewer's Responses to Questions

**Comments to the Author**

1. Is the manuscript technically sound, and do the data support the conclusions?

Reviewer #1: Yes

2. Has the statistical analysis been performed appropriately and rigorously?

Reviewer #1: Yes

3. Have the authors made all data underlying the findings in their manuscript fully available?

Reviewer #1: Yes

4. Is the manuscript presented in an intelligible fashion and written in standard English?

Reviewer #1: Yes

**Reviewer #1: ** Congratulations on studying prenatal attachment in adolescent and young adult women as well as its relationship with substance consumption.

The study is very interesting.

I believe the title should be more generic, and more for example:

Maternal-antenatal attachment in young pregnant women: social support, mentors, and fear of childbirth.

Should the title contain the specification of consumption? I ask you to reflect on this aspect.

**Do you want your identity to be public for this peer review?** For information about this choice, including consent withdrawal, please see our Privacy Policy

Reviewer #1: No

---

## [Author Response · Author response to Decision Letter 1]

19 May 2025

Thank you for the opportunity to revise our manuscript. To ensure that our manuscript meets PLOS ONE's style requirements, I have updated the author byline with appropriate indentation.

The funding statement was updated to include: “Research reported in this publication was supported by the National Institute On Drug Abuse of the National Institutes of Health under Award Number R01DA046401 (NDG). The funders had no role in study design, data collection and analysis, decision to publish, or preparation of the manuscript. There was no additional external funding received for this study.” As requested, this statement can be found in our cover letter.

Data for this analysis do not include any interview transcripts, only responses to an online survey and urine screening results. These data appear in a data repository as an Excel file and no personal details or potentially identifying information has been published. All data analyzed for this paper are identified with participant ID. No personal or identifiable information has been shared publicly.

A caption for my Supporting Information table (S1 Table) is included under the References section with the in-text citation updated to “S1 Table.”

The revised reference list is complete and correct. No cited papers have been retracted. The first author for reference 39 updated her last name, which is reflected in the revised reference list. I added middle initials for authors in reference 56. The order of references remained unchanged.

In response to the reviewer's comments, I edited the name of the article to reflect on more general aspects of investigation into maternal-antenatal attachment.

---

## [Editor Report · Decision Letter 1]

Maternal-antenatal attachment in young pregnant women: social support, mentors, and fear of childbirth

PONE-D-25-05398R1

Dear Dr. Alavala,

We’re pleased to inform you that your manuscript has been judged scientifically suitable for publication and will be formally accepted for publication once it meets all outstanding technical requirements.

Kind regards,

Wolfgang Blenau

Academic Editor

PLOS ONE
---

## [Editor Report · Acceptance letter]

PONE-D-25-05398R1

PLOS ONE

Dear Dr. Alavala,

I'm pleased to inform you that your manuscript has been deemed suitable for publication in PLOS ONE. Congratulations! Your manuscript is now being handed over to our production team.

Kind regards,

on behalf of

Dr. Wolfgang Blenau

Academic Editor

PLOS ONE